# P2Y_2_ and P2X4 Receptors Mediate Ca^2+^ Mobilization in DH82 Canine Macrophage Cells

**DOI:** 10.3390/ijms21228572

**Published:** 2020-11-13

**Authors:** Reece Andrew Sophocleous, Nicole Ashleigh Miles, Lezanne Ooi, Ronald Sluyter

**Affiliations:** 1Illawarra Health and Medical Research Institute, Wollongong, NSW 2522, Australia; rs256@uowmail.edu.au (R.A.S.); nam993@uowmail.edu.au (N.A.M.); lezanne@uow.edu.au (L.O.); 2Molecular Horizons and School of Chemistry and Molecular Bioscience, University of Wollongong, Wollongong, NSW 2522, Australia

**Keywords:** P2Y_2_ receptor, P2X4 receptor, canine, dog, purinergic signalling, DH82, macrophage, pain, neuroinflammation

## Abstract

Purinergic receptors of the P2 subclass are commonly found in human and rodent macrophages where they can be activated by adenosine 5′-triphosphate (ATP) or uridine 5′-triphosphate (UTP) to mediate Ca^2+^ mobilization, resulting in downstream signalling to promote inflammation and pain. However, little is understood regarding these receptors in canine macrophages. To establish a macrophage model of canine P2 receptor signalling, the expression of these receptors in the DH82 canine macrophage cell line was determined by reverse transcription polymerase chain reaction (RT-PCR) and immunocytochemistry. P2 receptor function in DH82 cells was pharmacologically characterised using nucleotide-induced measurements of Fura-2 AM-bound intracellular Ca^2+^. RT-PCR revealed predominant expression of P2X4 receptors, while immunocytochemistry confirmed predominant expression of P2Y_2_ receptors, with low levels of P2X4 receptor expression. ATP and UTP induced robust Ca^2+^ responses in the absence or presence of extracellular Ca^2+^. ATP-induced responses were only partially inhibited by the P2X4 receptor antagonists, 2′,3′-*O*-(2,4,6-trinitrophenyl)-ATP (TNP-ATP), paroxetine and 5-BDBD, but were strongly potentiated by ivermectin. UTP-induced responses were near completely inhibited by the P2Y_2_ receptor antagonists, suramin and AR-C118925. P2Y_2_ receptor-mediated Ca^2+^ mobilization was inhibited by U-73122 and 2-aminoethoxydiphenyl borate (2-APB), indicating P2Y_2_ receptor coupling to the phospholipase C and inositol triphosphate signal transduction pathway. Together this data demonstrates, for the first time, the expression of functional P2 receptors in DH82 canine macrophage cells and identifies a potential cell model for studying macrophage-mediated purinergic signalling in inflammation and pain in dogs.

## 1. Introduction

The activation of purinergic receptors by nucleotides such as adenosine 5′-triphosphate (ATP) and uridine 5′-triphosphate (UTP) is crucial for a number of inflammatory processes, including those in the central nervous system (CNS) such as chronic pain [1,2,3,4] and remyelination of nerves following injury to the CNS [5,6]. The P2 receptor family consists of seven mammalian ionotropic P2X receptors (P2X1-7) and eight mammalian metabotropic P2Y receptors (P2Y_1,2,4,6,11–14_) that can modulate intracellular Ca^2+^ concentrations through direct ion channel permeation or mobilization of intracellular Ca^2+^ stores, respectively [7]. P2 receptors, such as the P2X4 and P2Y_2_ receptors, are commonly expressed on human and rodent macrophages and macrophage cell lines [8,9,10,11,12,13,14], and have demonstrated roles in signalling pathways that control chronic pain and inflammation in humans or rodents [2,8,15,16,17,18]. Despite this, studies on purinergic signalling in canine macrophages are lacking. 

The DH82 cell line is a canine macrophage cell line isolated from a 10 year old Golden Retriever with malignant histiocytosis [19]. This cell line has recently been demonstrated as a useful model of canine macrophage physiology, bearing similarities to an M0 macrophage subtype with demonstrated potential for polarisation to either M1 or M2a subtypes through cytokine stimulation [20]. Studies have demonstrated that DH82 cells express a number of macrophage markers, such as CD11c and CD18 [21], and can secrete tumour necrosis factor (TNF)-α and interleukin (IL)-6 similar to that observed in lipopolysaccharide (LPS)-stimulated canine monocytes [22]. Despite its use as an in vitro model of viral and protozoan infection [23,24,25], knowledge regarding purinergic signalling in DH82 cells is limited to a single report describing ATP- and adenosine-induced cytokine release [26]. Although this study did not investigate any purinergic receptor per se, DH82 cells represent a possible model to study endogenous P2 receptors in canine macrophages for a number of reasons. Firstly, the original study revealed that ATP could alter cytokine expression in LPS-stimulated DH82 cells [26]. Secondly, infection of DH82 cells with canine distemper virus modulates inflammatory signalling pathways [27] that are common to P2X receptor-mediated signalling [28]. Thirdly, despite few studies analysing the expression of P2 receptors in dogs, it has been demonstrated that canine monocytes express P2X7 receptors [29,30]. Lastly, human and rodent macrophage or myeloid cell lines, such as THP-1 and RAW264.7 cells, are well-established models for studying endogenous P2 receptors commonly expressed on human and rodent macrophages [31,32,33,34,35,36,37,38].

The current study aimed to establish a canine macrophage model of P2 receptor signalling. Through investigation of canine P2 receptor expression and functional characterisation of these receptors, this study has identified the P2Y_2_ receptor and, to a lesser extent, the P2X4 receptor, as the primary functional P2 receptors in DH82 cells, which are responsible for nucleotide-mediated Ca^2+^ mobilization. 

## 2. Results

### 2.1. DH82 Cells Express Abundant P2RX4 mRNA Compared to Other P2 Receptors

To establish a P2 receptor mRNA expression profile for DH82 cells, cDNA was amplified by RT-PCR using primer pairs (Appendix A) designed to genes encoding canine P2X1-7 receptors and canine P2Y_1,2,4,6,11–14_ receptors and amplicons were semi-quantitatively analysed by agarose gel electrophoresis and densitometry. P2X4 receptor mRNA was most abundant in DH82 cells, with relative amounts comparable to glyceraldehyde 3-phosphate dehydrogenase (*GAPDH*; Figure 1) and β-actin (*ACTB*; data not shown). Other P2X receptor mRNAs, including P2X1 and P2X7 receptors were detected, but to a much lesser degree than the P2X4 receptor (Figure 1). Additionally, mRNA from a number of P2Y receptor subtypes were also detected, including P2Y_1_, P2Y_2_, P2Y_6_ and P2Y_11_ receptors, however, as with P2X1 and P2X7 receptors, these were expressed at greatly reduced levels compared to the P2X4 receptor (Figure 1).

### 2.2. Nucleotides Mediate Ca^2+^ Responses in DH82 Cells

To establish an agonist profile for functional P2 receptors in DH82 cells, nucleotides which have previously demonstrated activity towards P2 receptors from dogs, humans or rodents [39,40] were utilised to measure changes in intracellular Ca^2+^. These nucleotides were ATP, 3′-*O*-(4-benzoyl)benzoyl-ATP (BzATP), adenosine-5′-diphosphate (ADP), 2-methylthio-ADP (2MeSADP), uridine-5′-triphosphate (UTP) and uridine-5′-diphosphate (UDP). ADP was preincubated with hexokinase to remove trace amounts of ATP [41]. Incubation with ATP or UTP induced robust Ca^2+^ responses (ΔF340/380) in DH82 cells, which peaked approximately 15 s after application of nucleotides (Figure 2A,B). Ca^2+^ responses then decayed more slowly, returning to baseline approximately 70–80 s after the initial peak was observed (Figure 2A,B). Incubation with ADP or BzATP resulted in much smaller Ca^2+^ responses compared to ATP and UTP (Figure 2C,D). UDP and 2MeSADP were unable to induce Ca^2+^ responses up to 30 µM and 100 µM, respectively (Figure 2E,F). Decay time, calculated as the time constant (τ), was similar for ATP and UTP (τ = 56.9 ± 4.8 s and 56.7 ± 5.9 s, respectively), however, BzATP (τ = 102.3 ± 17.9 s) had a significantly longer decay time compared to ATP (*p* < 0.05), UTP (*p* < 0.05) or ADP (τ = 23.8 ± 0.4 s; *p* < 0.001), while UDP and 2MeSADP did not respond and as such, decay time could not be calculated (Figure 2G).

As Ca^2+^ responses were observed with a number of nucleotides, including those known to activate mammalian ionotropic P2X (ATP, BzATP) and metabotropic P2Y receptors (ATP, UTP, ADP), both the peak nucleotide-induced Ca^2+^ responses (Figure 2H) and net Ca^2+^ movement (Figure 2I; calculated as area under the curve [AUC]) were used for constructing concentration-response curves to account for potential differences in Ca^2+^ response phenotypes in a model of co-expression of P2X and P2Y receptor subtypes. In DH82 cells, nucleotides induced concentration-dependent Ca^2+^ responses with the rank order of potency of UTP > ATP >> ADP ≈ BzATP, with UDP and 2MeSADP being unresponsive (Figure 2H,I; Table 1). There were no significant differences between the EC_50_ values calculated for net Ca^2+^ movement and peak Ca^2+^ response for any nucleotide (Table 1).

### 2.3. P2X4 Receptors Mediate Minor Changes in Intracellular Ca^2+^ in DH82 Cells

#### 2.3.1. TNP-ATP Partially Reduces ATP-Induced Net Ca^2+^ Movement

To determine if the observed Ca^2+^ responses were mediated by P2X receptors, DH82 cells were preincubated with the non-selective P2X receptor antagonist, 2′,3′-*O*-(2,4,6-trinitrophenyl)-ATP (TNP-ATP) [42], then exposed to ATP. Preincubation with 50 µM TNP-ATP partially reduced Ca^2+^ responses mediated by 10 µM ATP, but did not significantly inhibit Ca^2+^ responses evoked by 1 µM or 100 µM ATP (Figure 3A–C). This was supported by a significant reduction in net Ca^2+^ movement at 10 µM ATP, but not at other ATP concentrations (Figure 3E). Despite this, no significant change in peak Ca^2+^ response or shift in decay time was observed (Figure 3D,F). Preincubation with TNP-ATP did not result in a significant shift in the peak Ca^2+^ response EC_50_ for ATP, compared to cells preincubated in the absence of TNP-ATP (Figure 3D; pEC_50_ 5.38 ± 0.08 vs. 5.57 ± 0.13, respectively; *p* = 0.133 Student’s *t*-test). In contrast, preincubation with TNP-ATP did result in a significant shift in the net Ca^2+^ movement EC_50_ compared to cells preincubated in absence of TNP-ATP (Figure 2E; pEC_50_ 5.04 ± 0.14 vs. 5.69 ± 0.13, respectively; *p* = 0.007 Student’s *t*-test).

#### 2.3.2. Paroxetine Partially Reduces ATP-Induced Net Ca^2+^ Movement

To further investigate the role of P2X receptors in DH82 cells, ATP-induced Ca^2+^ responses were measured in cells preincubated with paroxetine, a selective serotonin reuptake inhibitor which has been shown to inhibit P2X4 receptors [41,43,44] and human (but not rodent) P2X7 receptors [45,46]. Preincubation with paroxetine partially reduced Ca^2+^ responses mediated by ATP concentrations of 10 µM or greater, with a small, but non-significant inhibitory effect observed at 1 µM or below (Figure 4A–C). Similar to TNP-ATP, preincubation with paroxetine did not significantly reduce peak Ca^2+^ responses (Figure 4D), however, did significantly reduce net Ca^2+^ movement and decay kinetics (Figure 4E,F). Preincubation with paroxetine did not significantly shift the EC_50_ of ATP compared to cells preincubated in absence of paroxetine for peak Ca^+^ response (Figure 4D; pEC_50_ 5.55 ± 0.31 vs. 5.51 ± 0.48, respectively; *p* = 0.479 Student’s *t*-test) or net Ca^2+^ movement (Figure 4E; pEC_50_ 5.27 ± 0.35 vs. 5.4 ± 0.28, respectively; *p* = 0.387 Student’s *t*-test).

#### 2.3.3. 5-BDBD Partially Reduces ATP-Induced Net Ca^2+^ Movement

To determine further if P2X4 receptors played a role in the observed ATP-induced Ca^2+^ responses, DH82 cells were preincubated with the selective P2X4 receptor antagonist, 5-BDBD, of which was recently demonstrated to inhibit the canine P2X4 receptor [41]. Preincubation with 30 µM 5-BDBD had no significant inhibitory effects on the Ca^2+^ response mediated by ATP (Figure 5A–C). There was no significant difference in the EC_50_ for cells preincubated in the absence or presence of 5-BDBD for ATP-induced peak Ca^2+^ response (Figure 5D; pEC_50_ 5.87 ± 0.12 vs. 5.82 ± 0.10, respectively; *p* = 0.372 Student’s *t*-test) or net Ca^2+^ movement (Figure 5E; pEC_50_ 5.73 ± 0.18 vs. 5.64 ± 0.14, respectively; *p* = 0.358 Student’s *t*-test), although a trend towards decreased net Ca^2+^ movement was observed at 10 µM ATP (Figure 5E). Of note, there was a significant reduction in decay kinetics at 10 µM ATP for cells preincubated in the presence of 5-BDBD compared to those in absence of the antagonist (Figure 5F).

#### 2.3.4. Ivermectin Positively Modulates ATP-Induced Net Ca^2+^ Movement

It has recently been demonstrated that ivermectin can effectively potentiate Ca^2+^ responses mediated by canine P2X4 receptors [41]. To further investigate if DH82 cells express functional P2X4 receptors, cells were preincubated with ivermectin prior to activation with increasing concentrations of ATP. Preincubation with 3 µM ivermectin revealed a strong potentiation of ATP-induced Ca^2+^ responses in DH82 cells (Figure 6A–C), with significant increases in ATP-induced net Ca^2+^ movement and peak Ca^2+^ response observed in the presence of ivermectin compared to cells preincubated in absence of ivermectin at ATP concentrations upwards of 10 µM (Figure 6D,E). Despite this, there was no significant difference in the EC_50_ in the absence or presence of ivermectin for ATP-induced peak Ca^2+^ response (Figure 6D; pEC_50_ 5.76 ± 0.19 vs. 5.37 ± 0.35; *p* = 0.372 Student’s *t*–test) or net Ca^2+^ movement (Figure 6E; pEC_50_ 5.62 ± 0.01 vs. 5.57 ± 0.12, respectively; *p* = 0.358 Student’s *t*-test). Additionally, there were no significant differences in decay kinetics between cells preincubated in absence or presence of ivermectin (Figure 6F). Collectively this and the above data suggests that DH82 cells express functional P2X4 receptors.

### 2.4. ATP and UTP Mediate Both Ca^2+^ Influx and Store-Operated Ca^2+^ Entry in DH82 Cells

Despite the apparent P2X4-mediated effects on Ca^2+^ responses in DH82 cells, a lack of complete inhibition by P2X receptor antagonists, as well as the responsiveness to UTP, suggest the presence of functional P2Y receptors in DH82 cells. To determine if G_q/11_-coupled P2Y receptors, which have demonstrated roles in store-operated Ca^2+^ entry [47], were involved in the observed Ca^2+^ responses in DH82 cells, nucleotide-induced changes in intracellular Ca^2+^ were measured in the presence of extracellular or intracellular Ca^2+^ chelators. Compared to cells in the presence of extracellular Ca^2+^ (Figure 7A,F), cells incubated with ethylene glycol tetraacetic acid (EGTA) demonstrated a partial reduction in ATP- and UTP-induced Ca^2+^ responses (Figure 7B,G). This was supported by significant reductions in decay time (Figure 7K) and net Ca^2+^ movement (Figure 7L,M) for both ATP- and UTP-mediated responses. In contrast, preincubation with the cell-permeant Ca^2+^ chelator, 1,2-*bis*(2-aminophenoxy)ethane-*N*,*N*,*N*′,*N*′-tetraacetic acid tetrakis(acetoxymethyl ester) (BAPTA-AM), near completely reduced ATP- and UTP-induced Ca^2+^ responses (Figure 7C,H) and net Ca^2+^ movement (Figure 7L,M) in DH82 cells. Treatment with thapsigargin, to inhibit sarco/endoplasmic reticulum Ca^2+^ ATPase pumps [48], near completely reduced ATP- and UTP-induced Ca^2+^ responses in the presence of extracellular Ca^2+^ (ECS; Figure 7D,I,L,M) and completely reduced these responses in the absence of extracellular Ca^2+^ (EGTA; Figure 7E,J,L,M). Together, this suggests the involvement of both P2X and P2Y receptors in DH82 cells for mediating changes in intracellular Ca^2+^ in response to activation by nucleotides.

### 2.5. P2Y_2_ Receptor Activation Mediates Ca^2+^ Mobilization in DH82 Cells

#### 2.5.1. Suramin Reduces ATP- and UTP-Induced Ca^2+^ Mobilization

The data presented above suggests a major role for an ATP- and UTP-responsive P2Y receptor in Ca^2+^ mobilization within DH82 cells. Previous studies have demonstrated that the canine P2Y_2_ receptor in Madin–Darby canine kidney (MDCK) cells responds to both ATP and UTP with similar potency [49]. Therefore, to determine if nucleotide-induced Ca^2+^ mobilization was mediated by P2Y_2_ receptors, DH82 cells were preincubated with increasing concentrations of the non-selective P2 receptor antagonist, suramin [50], which is selective for P2Y_2_ over P2Y_4_ receptors [51]. Cells were then incubated with ATP or UTP at their respective EC_80_ to determine the optimal concentration for P2Y receptor inhibition. Preincubation of DH82 cells with 1 mM suramin inhibited Ca^2+^ responses evoked by 3 µM ATP, however lower concentrations of suramin (<100 µM) had little to no inhibitory effect (Figure 8A). In contrast, preincubation with 100 µM and 1 mM, but not 10 µM suramin or less inhibited Ca^2+^ responses evoked by 1 µM UTP (Figure 8B). Inhibitory effects observed in the presence of 100 µM suramin resulted in significant shifts in the IC_50_ of suramin between ATP- and UTP-induced peak Ca^2+^ responses (Figure 8C; pIC_50_ 3.02 ± 0.06 and 3.70 ± 0.05, respectively; *p* < 0.001 Student’s *t*-test) and net Ca^2+^ movement (Figure 8D; pIC_50_ 3.03 ± 0.12 and 3.54 ± 0.14, respectively; *p* = 0.025 Student’s *t*-test).

#### 2.5.2. AR-C118925 Reduces ATP- and UTP-Induced Ca^2+^ Mobilization

To determine if the P2Y_2_ receptor was mediating nucleotide-induced Ca^2+^ mobilization in DH82 cells, the selective P2Y_2_ receptor antagonist AR-C118925 [52] was preincubated with cells prior to incubation with ATP or UTP at their respective EC_80_ concentrations. AR-C118925 at concentrations of 1 µM or greater could only partially inhibit Ca^2+^ responses evoked by 3 µM ATP (Figure 8E). In contrast, preincubation with AR-C118925 at concentrations of 10 µM or greater near completely inhibited Ca^2+^ responses evoked by 1 µM UTP (Figure 8F). A significant shift was observed in the IC_50_ of AR-C118925 in response to activation by ATP and UTP calculated using peak Ca^2+^ responses (Figure 8G; pIC_50_ 6.18 ± 0.16 and 6.61 ± 0.09, respectively; *p* = 0.033 Student’s *t*-test), but not net Ca^2+^ movement (Figure 8H; pIC_50_ 6.67 ± 0.19 and 6.87 ± 0.12, respectively; *p* = 0.198 Student’s *t*-test). The inhibition of nucleotide-induced Ca^2+^ responses by AR-C118925 supports the presence of P2Y_2_ receptors in DH82 cells. Moreover, the differing effect of this antagonist on ATP- and UTP-induced responses indicates the presence of other P2 receptors in this cell line. Additionally, preincubation of DH82 cells together with 5-BDBD and AR-C118925 resulted in a complete inhibition of both ATP- and UTP-induced net Ca^2+^ movement (Appendix A), further suggesting a role for both P2X4 and P2Y_2_ receptors in nucleotide-mediated Ca^2+^ responses in DH82 cells. 

### 2.6. DH82 Canine Macrophages Predominantly Express Cell Surface P2Y_2_ Receptors

To confirm the presence of P2X4 and P2Y_2_ receptors, DH82 cells were analysed by immunocytochemistry and confocal microscopy using anti-P2X4 or anti-P2Y_2_ receptor antibodies. Confocal microscopy revealed the presence of both P2X4 and P2Y_2_ receptors in fixed and permeabilised DH82 cells (Figure 9A,B). The expression of P2X4 receptors was relatively low and largely intracellular (Figure 9A). The expression of P2Y_2_ receptors on DH82 cells was considerably higher and predominantly localised to the cell surface (Figure 9B), consistent with its reported expression in the membrane of MDCK cells [53,54]. No fluorescence was detected in DH82 cells stained with secondary antibodies alone (Appendix A). 

### 2.7. P2Y_2_ Receptor Activation Downstream Ca^2+^ Mobilization Is Coupled to the Phospholipase C/Inositol Triphosphate Signal Transduction Pathway in DH82 Cells

Functional P2Y_2_ receptors have been reported in human and rodent macrophages [9,10], where they can activate phospholipase C (PLC) and inositol trisphosphate (IP_3_) receptors, leading to Ca^2+^ mobilization from endoplasmic reticulum stores [8,12]. To determine if activation of canine P2Y_2_ receptors in DH82 macrophage cells results in a similar downstream signalling pathway, cells were preincubated with antagonists of PLC (U-73122) and IP_3_ receptors (2-aminoethoxydiphenyl borate; 2-APB), and nucleotide-induced intracellular Ca^2+^ mobilization was recorded in absence of extracellular Ca^2+^. The presence of AR-C118925 under these conditions was also examined. Preincubation of DH82 cells with AR-C118925 near completely inhibited intracellular Ca^2+^ mobilization mediated by ATP (Figure 10A; 90.5 ± 3.3% inhibition) or UTP (Figure 10B; 87.3 ± 7.4% inhibition). Similarly, preincubation of DH82 cells with 75 µM 2-APB near completely inhibited intracellular Ca^2+^ mobilization mediated by ATP (Figure 10A; 90.7 ± 4.4% inhibition) or UTP (Figure 10B; 91.2 ± 3.1% inhibition). In contrast, preincubation with 5 µM U-73122 only partially reduced intracellular Ca^2+^ mobilization mediated by ATP (Figure 10A; 64.9 ± 5.9% inhibition) or UTP (Figure 10B; 55.5 ± 8.7% inhibition). The combination of two or three of these antagonists completely impaired ATP- and UTP-induced Ca^2+^ responses (Figure 10A,B). Thus, preincubation with AR-C118925, 2-APB, U-73122 or any combination of these antagonists resulted in a significant reduction of intracellular Ca^2+^ mobilization (*p* < 0.001, one-way ANOVA) compared to that mediated by ATP (Figure 10A) or UTP (Figure 10B) in the absence of antagonists.

## 3. Discussion

To date, the DH82 canine macrophage cell line has primarily been utilized as a model of viral and protozoan infection [23,24,25], such as for the study of canine distemper and its oncolytic potential [55,56,57]. Although studies have described the expression of functional P2 receptors in human or rodent macrophages and macrophage cell lines [8,10,11,34], studies directly investigating P2 receptors in canine macrophages have been absent. The current study described, for the first time, the expression and function of P2 receptors in DH82 cells, demonstrating a primary role for cell surface P2Y_2_ receptors in nucleotide-mediated Ca^2+^ mobilization through PLC/IP_3_ signal transduction. This study also demonstrates a minor functional role for P2X4 receptors in DH82 cells, suggesting this cell line may present as a suitable model for studying P2 receptor-mediated inflammation and pain signalling in dogs.

The agonist profile of ATP on DH82 cells demonstrated pharmacological similarities to the recombinant canine P2X4 receptor [41], as well as studies of endogenous P2X4 receptors in a human macrophage cell model [34]. BzATP induced partial Ca^2+^ responses in DH82 cells with significantly lower potency compared to ATP, consistent with the recent report that BzATP is a partial agonist of recombinant canine P2X4 receptors [41]. In addition, the increased decay kinetics of Ca^2+^ responses evoked by BzATP, compared to ATP, further supports a role for P2X4 receptors in the observed Ca^2+^ responses [41]. TNP-ATP and paroxetine, two non-selective antagonists of P2X4 receptors [44], as well as 5-BDBD, a selective P2X4 receptor antagonist [58], had minor inhibitory effects on ATP-induced Ca^2+^ responses in DH82 cells. Although it has recently been demonstrated that these antagonists can inhibit recombinant canine P2X4 receptors, the minor inhibition observed with these antagonists in DH82 cells were potentially in part due a lack of potency towards the canine P2X4 receptor [41], as well as the relatively low expression of P2X4 receptors in DH82 cells observed by immunocytochemistry. In contrast, ivermectin, the positive allosteric modulator which is routinely used to investigate P2X4 receptor activity [59], demonstrated strong potentiation of ATP-induced Ca^2+^ responses and efficacy of ATP, with little effect on decay time. This data further supports the expression of functional P2X4 receptors in canine macrophages, however, it suggests that potentiation or upregulation of P2X4 receptors may first be required to observe notable responses.

Consistent with the pharmacological profiles reported for the canine P2Y_2_ receptor cloned from MDCK cells [49], both ATP and UTP induced robust Ca^2+^ responses in DH82 cells with similar EC_50_ values. These responses were observed even in the absence of extracellular Ca^2+^, consistent with the P2Y-mediated mobilization of intracellular Ca^2+^ [60]. Similar to the current study with DH82 cells, other studies have also demonstrated that ADP is a low-potency agonist of the canine P2Y_2_ receptor cloned from MDCK cells [49,61]. In addition, incubation of DH82 cells with BzATP revealed pharmacological similarities to that observed with human P2Y_2_ receptors, where BzATP is ineffective at concentrations below 100 µM [33,62]. Collectively, this suggests that functional P2Y_2_ receptors were responsible for nucleotide-induced Ca^2+^ mobilization in DH82 cells. This was supported by inhibition observed in the presence of the P2Y_2_ receptor antagonists, suramin and AR-C118925. Despite suramin lacking potency and selectivity, it remains a valuable tool in characterising P2Y receptor responses, as it is considered a low-potency antagonist of P2Y_2_ receptors, but is relatively insensitive to P2Y_4_ receptors [51,63]. This suggests that the ATP/UTP-sensitive P2Y_2_, but not P2Y_4_ receptor, is responsible for the observed Ca^2+^ mobilization, consistent with the expression of P2Y_2_ receptors in DH82 cells determined by confocal microscopy. Notably, P2Y_2_ receptor protein expression was greater than that of P2X4 receptor protein expression, with an opposite pattern observed for mRNA expression of these receptors. Reasons for this discrepancy remain unknown, but a lack of correlation between mRNA and protein expression is well documented and attributed to various contributing factors related to post-transcriptional and post-translational regulation of mRNA and protein expression [64].

A number of other canine P2Y receptors from MDCK cells have been cloned and characterised, including the P2Y_1_ and P2Y_11_ receptors [49,65,66], of which mRNA of both these receptors were detected in DH82 cells. The nucleotide agonist profile of canine P2Y_11_ receptors differs markedly from the human P2Y_11_ receptor, where ATP is a potent agonist of human, but not canine P2Y_11_ receptors, and ADP and its analogue 2MeSADP are potent agonists of canine, but not human P2Y_11_ receptors [66]. In the current study, it was revealed that ATP, but not ADP, was a moderately potent mediator of Ca^2+^ responses in DH82 cells, while no such responses were observed with 2MeSADP. BzATP is also a full agonist of human P2Y_11_ receptors [67], further suggesting that DH82 cells likely do not express functional P2Y_11_ receptors. ADP and 2MeSADP have also been reported as potent agonists of the canine P2Y_1_ receptor [68]. However, it was demonstrated in the current study that only ADP, but not 2MeSADP, induced a small Ca^2+^ response in DH82 cells. Although this could suggest that DH82 cells express low amounts of functional P2Y_1_ receptors, the complete lack of response to 2MeSADP suggests that P2Y_1_ receptors are unlikely to be responsible for P2Y receptor-mediated Ca^2+^ mobilisation in DH82 cells. Additionally, a complete lack of Ca^2+^ response in DH82 cells incubated with the P2Y_6_ receptor agonist, UDP [69,70], strongly suggests that DH82 cells do not express functional P2Y_6_ receptors.

The current study demonstrated that nucleotide-mediated Ca^2+^ mobilization in DH82 cells was also inhibited by antagonists of PLC and IP_3_ receptors, U-73122 and 2-APB, respectively. This was consistent with previous studies that demonstrate coupling of P2Y_2_ receptors to G_q/11_ and downstream signalling pathways in MDCK cells [49,65,71]. While 2-APB near completely inhibited Ca^2+^ mobilisation, U-73122 only resulted in partial inhibition, although higher concentrations (>10 µM) have been shown to completely block P2Y_2_ receptor-mediated Ca^2+^ responses [72]. Notably, pre-incubation with P2Y_2_ receptor antagonists resulted in approximately two-fold greater inhibition of UTP-induced Ca^2+^ responses compared to ATP-induced responses, suggesting that ATP remained active at other receptors involved in mediating changes in intracellular Ca^2+^, such as P2X4 receptors, which are also relatively insensitive to suramin [73]. In addition, both ATP- and UTP-induced Ca^2+^ responses could be completely inhibited by co-incubation with 5-BDBD and AR-C118925, supporting a role for both P2X4 and P2Y_2_ receptors in DH82 cells.

Pro-monocytic and macrophage-like cell lines, such as human THP-1 cells, have recently proven useful models for studying endogenous purinergic signalling via P2X4 and P2Y_2_ receptors [10,13,34]. However, studies have demonstrated that these cell lines can be polarised towards a more specialised macrophage phenotype, in which the expression of P2 receptors, such as P2X4 and P2X7 receptors, are commonly upregulated [34,38,74]. A study has recently demonstrated that DH82 canine macrophage cells could be polarised towards the M1 or M2a subtype through cytokine stimulation [20]. However, the DH82 cells utilised throughout this study remained unpolarised (M0) and, thus, it remains to be determined if cytokine stimulation influences P2 receptor expression or function. To this end, future studies could determine the purinergic signalling landscape of polarised DH82 cells. Future studies could also analyse P2 receptor expression and signalling in native canine macrophages. Given the known expression of P2X7 receptors on canine monocytes [29,30], canine monocyte-derived macrophages may provide a suitable candidate for the study of other purinergic receptors in native canine macrophages.

The upregulation of P2X4 receptors in macrophages and microglia has been highlighted as a key component in the signalling of inflammatory conditions, including chronic inflammatory and neuropathic pain [75,76], and remyelination of damaged nerves in the CNS [77]. P2X4 receptors, which reside primarily within lysosomes of macrophages, can be upregulated at the cell surface through lysosomal exocytosis [78]. This process plays a key role in Ca^2+^ homeostasis, ATP release and local activation of cell surface purinergic receptors [79]. Notably, activation of the C-C chemokine receptor 2 (CCR2) by C-C chemokine ligand 2 (CCL2) is known to mediate lysosomal exocytosis [80], while in rat alveolar macrophages and human THP-1 cells it has been demonstrated that the activation of cell surface P2Y_2_ receptors induces the upregulation and secretion of CCL2 [8,10]. Thus, it could be suggested that activation of macrophage P2Y_2_ receptors results in a P2 receptor signalling feedback mechanism which results in an increase in Ca^2+^ flux through upregulation of lysosomal exocytosis and trafficking of P2X4 receptors to the cell surface, leading to the release of prostaglandin E_2_ and subsequent chronic inflammatory pain signalling [2]. These activated macrophages may also modulate microglial P2X4 receptors to control neuroinflammatory signalling following injury to the central nervous system [81]. Given the expression profile of these receptors in DH82 canine macrophages, this cell line may provide a suitable model for studying inflammatory pain signalling mechanisms of dogs in vitro, as well as for the pre-clinical testing of novel therapeutics targeting chronic pain.

Finally, although the sequence of P2X4 and P2Y_2_ receptors in DH82 cells is yet to be determined, it remains of interest to identify novel single nucleotide polymorphisms should they exist in the genes encoding these receptors in dogs. A recent whole genome study of 582 dogs has revealed at least one missense variant (Ala9Asp) within the canine *P2RX4* gene and two missense variants (Gly193Ser and Val375Ile) within the canine *P2RY2* gene [82] (data accessed from the European Variation Archive; https://www.ebi.ac.uk/eva/). Whilst the effects of single nucleotide polymorphisms in the canine *P2RX4* and *P2RY2* genes are largely unknown, it has been demonstrated that single nucleotide polymorphisms of the genes encoding the human P2X4 and P2Y_2_ receptors can alter receptor function [83,84]. Notably, in human macrophages, a 312Ser polymorphism of the P2Y_2_ receptor has been demonstrated to alter secretion of CCL_2_ following activation by UTP [10], suggesting a potential association with macrophage-mediated chronic inflammatory pain signalling. Despite this however, studies by our group have demonstrated that the canine *P2RX4* and *P2RX7* genes are much more conserved than their human counterparts [41,85,86] and, as such, naturally-occurring polymorphisms in the canine *P2RY2* gene may also be rare or limited to uncommon breeds not frequently sampled in canine whole genome studies.

In conclusion, the current study demonstrates for the first time, that DH82 canine macrophages primarily express functional P2Y_2_ receptors and low levels of functional P2X4 receptors. As such, DH82 cells provide the first canine macrophage cell line for the study of endogenous P2X4 and P2Y_2_ receptors. The data presented here provides indirect evidence that P2X4 and P2Y_2_ receptors play a role in mediating changes in intracellular Ca^2+^ in canine macrophages in vivo. This mimics events observed in human and rodent macrophages and macrophage cell lines, where these P2 receptors have been suggested to play a key role in inflammation and chronic pain. Thus, DH82 cells may aid in the study of P2 receptor-mediated inflammation, including neuroinflammatory signalling processes, as well as preclinical screening of novel P2 receptor-targeting compounds for potential use in the treatment of inflammatory conditions, such as chronic pain in dogs.

## 4. Materials and Methods 

### 4.1. Compounds and Reagents

BSA, EGTA and reagents for producing Ca^2+^ solutions were from Amresco (Solon, OH, USA). Fetal bovine serum (FBS) was purchased from Bovogen Biologicals (East Keilor, Melbourne, Australia) and heat inactivated at 56 °C for 30 min before use. 2-APB, U-73122 and UDP were from Cayman Chemical (Ann Arbor, MI, USA). Primers for RT-PCR were from Integrated DNA Technologies (Coralville, IA, USA). 5-BDBD, ADP (pre-treated with hexokinase as per [41]), ATP, BAPTA-AM, BzATP, hexokinase from *Saccharomyces cerevisae*, ivermectin, MEM non-essential amino acid solution, paraformaldehyde, paroxetine, phosphate buffered saline (PBS), poly-_D_-lysine hydrobromide (5 µg∙mL^−1^ working stock), pluronic F-127, saponin, suramin and UTP were from Sigma-Aldrich (St. Louis, MO, USA). DMEM/F12 medium, ExoSAP-IT, Fura-2 AM, GlutaMAX, penicillin-streptomycin and 0.05% trypsin-EDTA were from ThermoFisher Scientific (Melbourne, Australia). 2MeSADP, AR-C118925, thapsigargin and TNP-ATP were from Tocris Bioscience (Bristol, UK).

### 4.2. Cells

DH82 cells were obtained from the European Collection of Authenticated Cell Cultures (ECACC cat. no. 94062922, RRID: CVCL_2018). DH82 cells were cultured in DMEM/F12 medium supplemented with 10% FBS, 2 mM GlutaMAX, 100 U/mL penicillin, 100 µg/mL streptomycin and 1% non-essential amino acids at 37 °C/5% CO_2_. Cells were routinely found to be negative for mycoplasma contamination using the MycoAlert Mycoplasma Detection Kit (Lonza, Waverley, Australia).

### 4.3. RNA Isolation, cDNA Synthesis and RT-PCR

Total RNA was extracted from DH82 cells using the ISOLATE II RNA Mini Kit (Bioline, London, UK) according to manufacturer’s instructions. cDNA was synthesised from RNA using the qScript cDNA SuperMix Kit (Quanta Biosciences, Gaithersburg, MD, USA) according to manufacturer’s instructions. RT-PCR amplification of cDNA was carried out using the primer pairs and conditions listed in Appendix A, the MangoTaq DNA polymerase kit (Bioline) and a Mastercycler Pro S (Eppendorf, Hamburg, Germany). PCR cycling consisted of initial denaturation at 95 °C for 2 min, followed by 35 cycles of denaturation at 95 °C for 30 s, annealing at 49–61 °C for 30 s and extension at 72 °C for 1 min. Amplicons were treated with ExoSAP-IT and loaded onto a 1% agarose gel and imaged using GelRed Nucleic Acid Gel Stain (Biotium, Fremont, CA, USA) and a Bio-Rad Molecular Imager Gel Doc XR+ (Hercules, CA, USA). Densitometry quantification was carried out using ImageJ [87] analysis software.

### 4.4. Measurement of Intracellular Ca^2+^

Measurements of intracellular Ca^2+^ were determined using Fura-2 AM as previously described [41]. Recordings were performed in extracellular Ca^2+^ solution (ECS; 145 mM NaCl, 2 mM CaCl_2_, 1 mM MgCl_2_, 5 mM KCl, 13 mM glucose and 10 mM HEPES, pH 7.4) or in Ca^2+^ free solution (145 mM NaCl, 1 mM MgCl_2_, 5 mM KCl, 13 mM glucose and 10 mM HEPES, pH 7.4) for recordings in absence of Ca^2+^. Cells were plated at 6 × 10^4^ cells/well in poly-_D_-lysine-coated black-walled µClear bottom 96-well plates (Greiner Bio-One, Frickenheisen, Germany) and incubated at 37 °C/5% CO_2_ for 18–24 h. Cells were washed in ECS then preincubated with Fura-2 AM loading buffer (2.5 µM Fura-2 AM/0.2% pluronic acid in ECS) in the dark for 30 min at 37 °C. Prior to recording fluorescence, excess Fura-2 was removed and cells were washed with ECS (or Ca^2+^ free solution for recordings in absence of Ca^2+^), then incubated for a further 20 min to allow for complete de-esterification. Fura-2 fluorescence emission at 510 nm was recorded every 5 s at 37 °C using a Flexstation3 (Molecular Devices, Sunnyvale, CA, USA) following excitation at 340 and 380 nm. Recordings were taken for 15 s prior to addition of compounds to establish baseline fluorescence then for 3–5 min after addition of agonists. Where indicated, cells were preincubated with antagonists for up to 30 min prior to addition of nucleotides. The relative change in intracellular Ca^2+^ (ΔCa^2+^) was calculated as ratio of Fura-2 fluorescence following excitation at 340 nm and 380 nm (F_340/380_) was determined and normalized to the mean basal fluorescence according to the formula (1):(1)$$\Delta {\mathrm{Ca}}^{2+}\;=\frac{\Delta F}{F}=\;\frac{F-F_{\mathrm{rest}}}{F_{\mathrm{rest}}}$$
where F is the F_340/380_ at any given time and F_rest_ is the mean fluorescence of the given well prior to the addition of nucleotides [88]. To investigate endogenous P2X and P2Y receptor-mediated Ca^2+^ responses in DH82 canine macrophages, both the peak Ca^2+^ response (F_340/380_) and the net Ca^2+^ movement (calculated as area under the curve using GraphPad Prism) were calculated and, where indicated, used for fitting concentration-response curves fit to the Hill equation using the least squares method. Decay time (τ, time constant) was calculated from the peak F_340/380_ using the nonlinear regression one phase decay model for GraphPad Prism. Decay times were not calculated where no response to agonists was recorded. Where antagonist IC_50_ was calculated against the approximate EC_80_ of ATP or UTP, responses were normalized to the response in absence of antagonist to allow data to be fit to a curve.

### 4.5. Immunocytochemistry and Confocal Microscopy

Cells were plated at 1 × 10^5^ cells/18 mm glass coverslip in 24-well plates (Greiner Bio-One) and incubated at 37 °C/5% CO_2_ overnight prior to use. Cells were fixed with 3% (*w*/*v*) paraformaldehyde at 4 °C for 15 min then washed three times with PBS. Cells were permeabilized with 0.1% (*w*/*v*) saponin resuspended in a blocking buffer (2% BSA (*w*/*v*) in PBS) at room temperature for 15 min and then incubated with anti-P2Y_2_ (1:250; Alomone, cat no. APR-102, RRID: AB_2756769) or anti-P2X4 (1:250; Sigma-Aldrich cat no. SAB2500734, RRID:AB_10604119) primary antibody in 2% BSA/PBS at room temperature for 2 h. Cells were washed three times with PBS and then incubated with Alexa Fluor594-conjugated anti-rabbit (1:200; Abcam cat no. ab150080, RRID: AB_2650602), or Alexa Fluor594-conjugated anti-goat (1:200; Abcam cat no. ab150136, RRID: AB_2782994) secondary antibody in 2% BSA/PBS at room temperature for 60 min. Cells were washed three times with PBS and then incubated with secondary antibody in 2% BSA/PBS at room temperature for 60 min. Washed coverslips were mounted onto a glass slide using 50% glycerol in PBS and sealed with nail polish. Cells were visualized on a Leica (Mannheim, Germany) SP5 confocal microscope.

### 4.6. Data and Statistical Analysis

All data were analysed using GraphPad Prism 5. Half-maximal effective and inhibitory concentrations (EC_50_ and IC_50_, respectively) are expressed as their negative logarithm (pEC_50_/pIC_50_) ± SEM. Data were compared using a two-tailed Student’s t test or one-way ANOVA with Bonferroni post hoc test for single or multiple comparisons, respectively. Multiple comparisons involving two interdependent variables were analysed using a two-way ANOVA with Bonferroni post hoc test. Throughout this study *p* < 0.05 was considered statistically significant.

### 4.7. Nomenclature of Targets and Ligands

All targets and ligands used throughout this manuscript conform with the guidelines outlined by the International Union of Basic and Clinical Pharmacology and British Pharmacological Society (IUPHAR/BPS) Guide to Pharmacology [40,89].

## Figures and Tables

**Figure 1 ijms-21-08572-f001:**
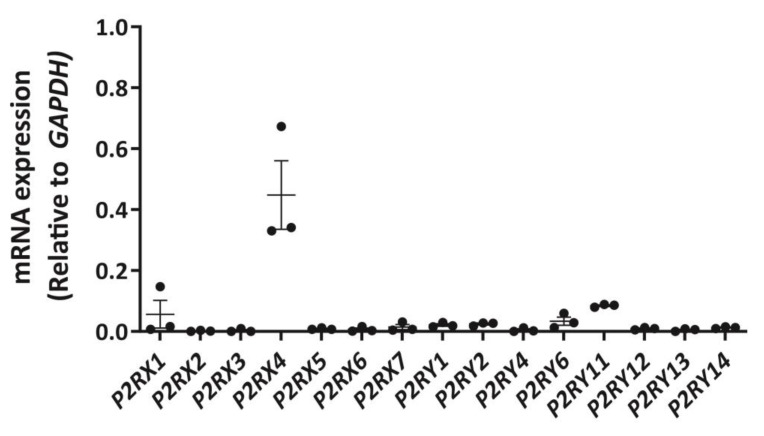
Expression of P2X and P2Y receptor mRNA in DH82 cells. RNA was isolated from DH82 cells and cDNA was synthesized and amplified using primer pairs designed to each respective *P2RX* or *P2RY* genes, with glyceraldehyde 3-phosphate dehydrogenase (*GAPDH*) as a positive control. Amplification in absence of cDNA was conducted for each primer pair to ensure primer specificity. Amplicons were visualized by agarose gel electrophoresis using GelRed and the GelDoc XR+ imaging system and semi-quantitatively analysed by densitometry. Data shown are the mean ± SEM relative to *GAPDH* expression from three independent experiments.

**Figure 2 ijms-21-08572-f002:**
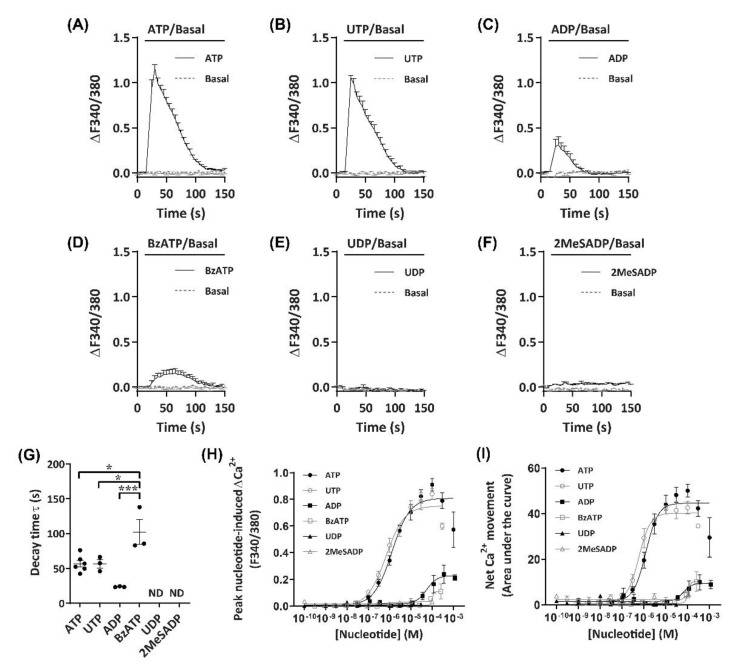
Nucleotide-induced Ca^2+^ response profiles for DH82 cells. (**A**–**I**) DH82 cells in extracellular Ca^2+^ solution (ECS) were loaded with Fura-2, incubated in the absence (basal) or presence of each nucleotide (as indicated) and Fura-2 fluorescence was recorded. Ca^2+^ traces (ΔF340/380) for (**A**) 100 µM adenosine 5′-triphosphate (ATP) (*n* = 6), (**B**) 30 µM uridine 5′-triphosphate (UTP) (*n* = 3), (**C**) 100 µM adenosine 5′-diphosphate (ADP) (preincubated with 4.5 U/mL hexokinase for 1 h at 37 °C) (*n* = 3), (**D**) 300 µM 3’-*O*-(4-benzoyl)benzoyl-ATP (BzATP) (*n* = 3), (**E**) 30 µM uridine 5′-diphosphate (UDP) (*n* = 3) and (**F**) 100 µM 2-methylthio-ADP (2MeSADP) (*n* = 3). (**G**) One phase decay time (τ) calculated from the peak of each Ca^2+^ trace, ND = no data. (**H**) Peak nucleotide-induced Ca^2+^ responses and (**I**) net Ca^2+^ movement were fit to the Hill equation to produce concentration-response curves (*n* values correspond to respective individual traces above). (**A**–**I**) Data shown are the mean ± SEM from three to six independent experiments as indicated. (**G**) * *p* < 0.05 and *** *p* < 0.001 between nucleotides as indicated analysed using a one-way ANOVA with Bonferroni post hoc test.

**Figure 3 ijms-21-08572-f003:**
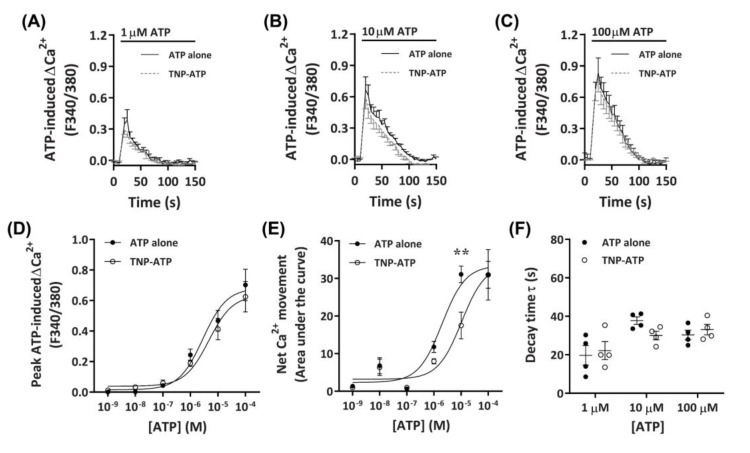
ATP-induced Ca^2+^ responses in DH82 cells in the absence or presence of 2′,3′-*O*-(2,4,6-trinitrophenyl)-ATP (TNP-ATP). (**A–F**) DH82 cells in ECS were loaded with Fura-2 and preincubated in the absence (ATP alone) or presence of 50 µM TNP-ATP (in ECS) for 5 min. Cells were exposed to increasing concentrations of ATP and Fura-2 fluorescence was recorded. (**A**–**C**) ATP-induced Ca^2+^ traces (F340/380) were plotted and the (**D**) peak Ca^2+^ response and (**E**) net Ca^2+^ movement were fit to the Hill equation to produce concentration-response curves. (**F**) One phase decay time (τ) calculated from the peak of each Ca^2+^ trace in (**A**–**C**). (**A**–**F**) Data shown are the mean ± SEM from four independent experiments. (**D**–**F**) ** *p* < 0.01 compared to respective concentration of ATP alone analysed using a two-way ANOVA with Bonferroni post hoc test.

**Figure 4 ijms-21-08572-f004:**
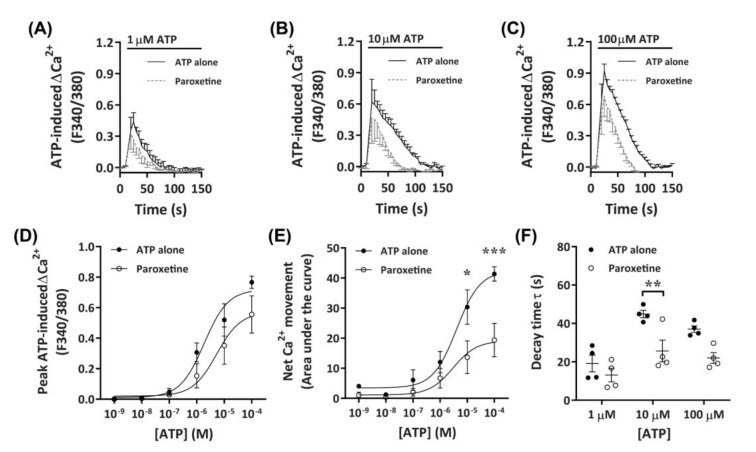
ATP-induced Ca^2+^ responses in DH82 cells in the absence or presence of paroxetine. (**A**–**F**) DH82 cells in ECS were loaded with Fura-2 and preincubated in the absence (ATP alone) or presence of 100 µM paroxetine (both 0.3% dimethyl sulfoxide; DMSO) for 5 min. Cells were then exposed to increasing concentrations of ATP and Fura-2 fluorescence was recorded. (**A**–**C**) ATP-induced Ca^2+^ traces (F340/380) were plotted and the (**D**) peak Ca^2+^ response and (**E**) net Ca^2+^ movement were fit to the Hill equation to produce concentration-response curves. (**F**) One phase decay time (τ) calculated from the peak of each Ca^2+^ trace in (**A**–**C**). (**A**–**F**) Data shown are the mean ± SEM from four independent experiments. (**D**–**F**) * *p* < 0.05, ** *p* < 0.01 and *** *p* < 0.001 compared to respective concentration of ATP alone analysed using a two-way ANOVA with Bonferroni post hoc test.

**Figure 5 ijms-21-08572-f005:**
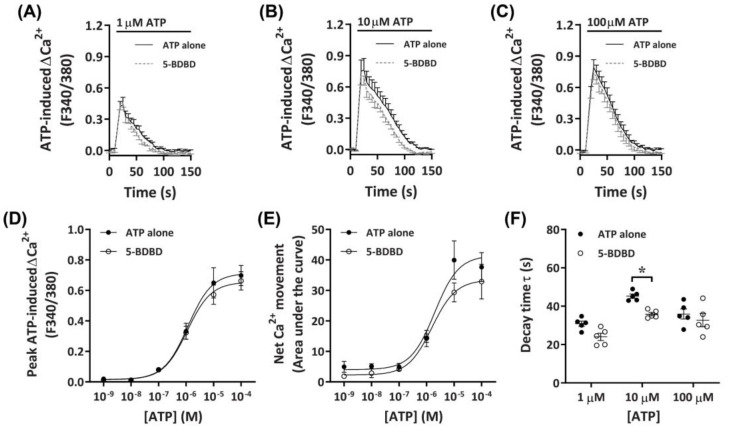
ATP-induced Ca^2+^ responses in DH82 cells in the absence or presence of 5-BDBD. (**A**–**F**) DH82 cells in ECS were loaded with Fura-2 and preincubated in the absence (ATP alone) or presence of 30 µM 5-BDBD (both 0.3% DMSO) for 5 min. Cells were then exposed to increasing concentrations of ATP and Fura-2 fluorescence was recorded. (**A**–**C**) ATP-induced Ca^2+^ traces (F340/380) were plotted and the (**D**) peak Ca^2+^ response and (**E**) net Ca^2+^ movement were fit to the Hill equation to produce concentration-response curves. (**F**) One phase decay time (τ) calculated from the peak of each Ca^2+^ trace in (**A**–**C**). (**A**–**F**) Data shown are mean ± SEM from five independent experiments. (**D**–**F**) * *p* < 0.05 compared to respective concentration of ATP alone analysed using a two-way ANOVA with Bonferroni post hoc test.

**Figure 6 ijms-21-08572-f006:**
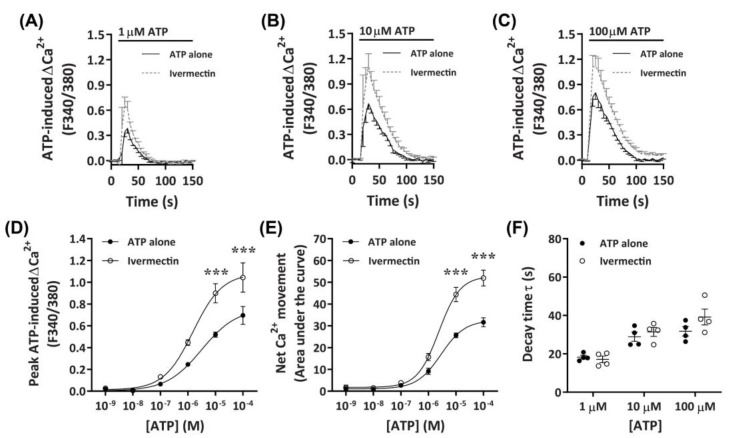
ATP-induced Ca^2+^ responses in DH82 cells in the absence or presence of ivermectin. (**A**–**F**) DH82 cells in ECS were loaded with Fura-2 and preincubated in the absence (ATP alone) or presence of 3 µM ivermectin (both 0.1% DMSO) for 5 min. Cells were then exposed to increasing concentrations of ATP and Fura-2 fluorescence was recorded. (**A**–**C**) ATP-induced Ca^2+^ traces (F340/380) were plotted and the (**D**) peak Ca^2+^ response and (**E**) net Ca^2+^ movement were fit to the Hill equation to produce concentration-response curves. (**F**) One phase decay time (τ) calculated from the peak of each Ca^2+^ trace in (**A**–**C**). (**A**–**F**) Data shown are mean ± SEM from five independent experiments. (**D**–**F**) *** *p* < 0.001 compared to respective concentration of ATP alone analysed using a two-way ANOVA with Bonferroni post hoc test.

**Figure 7 ijms-21-08572-f007:**
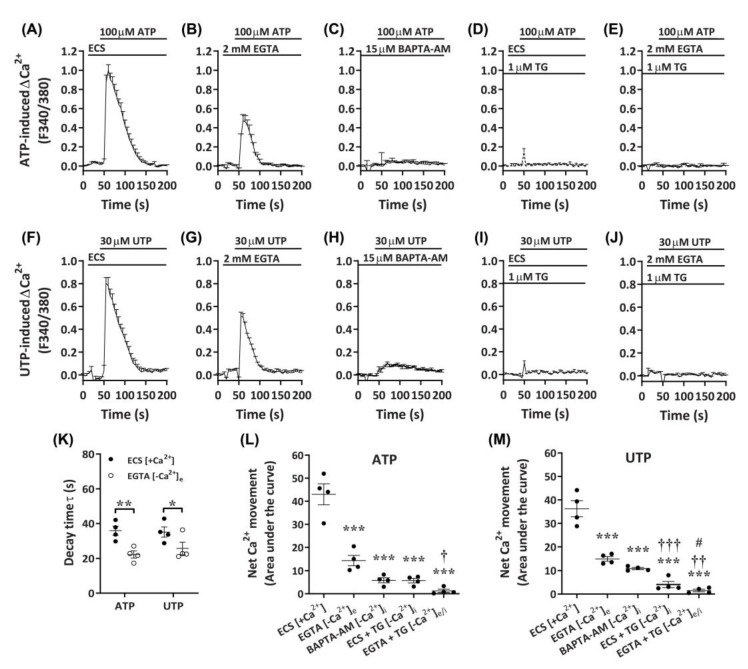
Nucleotide-induced Ca^2+^ responses in the absence or presence of extracellular and/or intracellular Ca^2+^. (**A**–**M**) DH82 cells in (**A**,**D**,**F**,**I**) ECS or (**B**,**C**,**E**,**G**,**H**,**J**) Ca^2+^-free solution were loaded with Fura-2 and preincubated in the absence (**A**,**D**,**F**,**I**) (ECS) or presence (**B**,**E**,**F**,**J**) of 2 mM ethylene glycol tetraacetic acid EGTA for 30 s, (**C**,**H**) 15 µM 1,2-*bis*(2-aminophenoxy)ethane-*N*,*N*,*N*′,*N*′-tetraacetic acid tetrakis(acetoxymethyl ester) (BAPTA-AM) for 30 min or (**D**,**E**,**I**,**J**) 1 µM thapsigargin (TG) for 30 min prior to incubation in the absence (**D**,**I**) or presence (**E**,**J**) of 2 mM EGTA. (**A**–**M**) Cells were then exposed to (**A**–**E**) 100 µM ATP or (**F**–**J**) 30 µM UTP and Fura-2 fluorescence was recorded. (**K**) One phase decay time (τ) calculated from the peak of each Ca^2+^ trace in (**A**,**B**,**F**,**G**). (**L**,**M**) Net Ca^2+^ movement from each trace in **A**–**J**. (**A**–**M**) Data shown are the mean ± SEM from four independent experiments. * *p* < 0.05, ** *p* < 0.01 and *** *p* < 0.001 compared to respective ECS alone control; ^†^
*p* < 0.05, ^††^
*p* < 0.01 and ^†††^
*p* < 0.001 compared to EGTA; ^#^
*p* < 0.05 compared to BAPTA-AM analysed using a (**K**) Student’s *t*-test or (**L**,**M**) one-way ANOVA with Bonferroni post hoc test.

**Figure 8 ijms-21-08572-f008:**
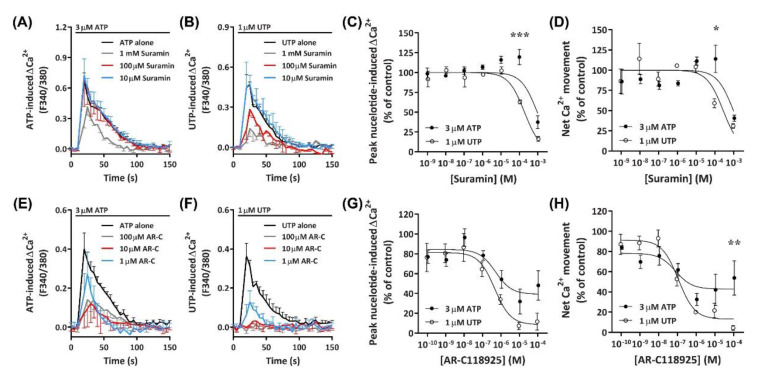
ATP- and UTP-induced Ca^2+^ responses in DH82 cells in the absence or presence of suramin or ARC-118925. (**A**–**H**) DH82 cells in ECS were loaded with Fura-2 and preincubated in the absence or presence of increasing concentrations of (**A**–**D**) suramin (in ECS) or (**E**–**H**) AR-C118925 (AR-C; 0.3% DMSO) for 30 min. Cells were then exposed to 3 µM ATP or 1 µM UTP (respective EC_80_ values) and Fura-2 fluorescence was recorded. Nucleotide-induced (**C**,**G**) peak Ca^2+^ response and (**D**,**H**) net Ca^2+^ movement were normalised to 3 µM ATP or 1 µM UTP alone and expressed as a percentage of the response in absence of inhibitor (% of control). Data were then to fit data to the Hill equation to produce concentration-response curves and calculate the IC_50_. (**A**–**H**) Data shown are mean ± SEM from three independent experiments. * *p* < 0.05, ** *p* < 0.01 and *** *p* < 0.001 compared to respective concentration of antagonist with ATP or UTP analysed using a two-way ANOVA with Bonferroni post hoc test.

**Figure 9 ijms-21-08572-f009:**
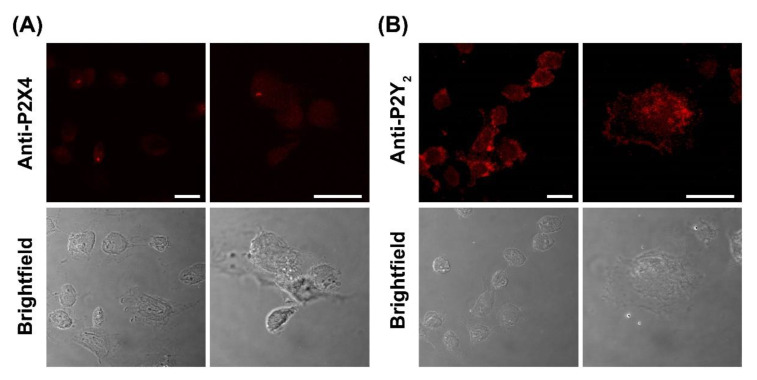
Expression of P2X4 and P2Y_2_ receptors in DH82 cells. DH82 cells were fixed, permeabilized and labelled with (**A**) anti-P2X4 or (**B**) anti-P2Y_2_ receptor primary antibodies, then with anti-goat^594^ or anti-rabbit^594^ secondary antibodies, respectively. Cells were imaged by confocal microscopy. Scale bar = 20 µm. Results are representative of three independent experiments.

**Figure 10 ijms-21-08572-f010:**
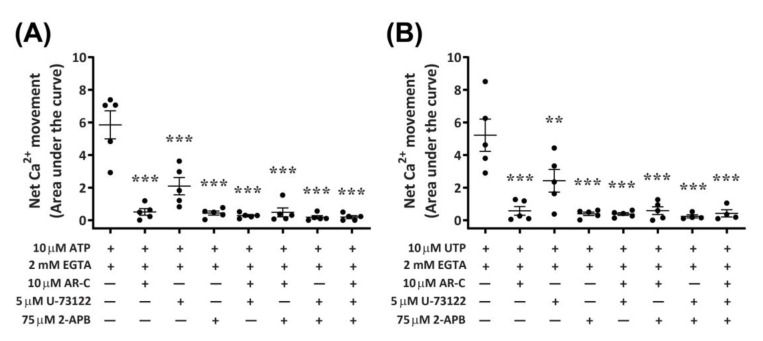
Nucleotide-induced Ca^2+^ mobilization in DH82 cells depleted of extracellular Ca^2+^ in the absence or presence of AR-C118925, U-73122 and 2-APB. DH82 cells were loaded with Fura-2 and preincubated in Ca^2+^-free solution containing 2 mM EGTA in the absence or presence of 10 µM AR-C118925 (AR-C; 0.03% DMSO), 5 µM U-73122 (0.05% DMSO) or 75 µM 2-APB (0.15% DMSO) for 5 min. Cells were then exposed to (**A**) 10 µM ATP or (**B**) 10 µM UTP and Fura-2 fluorescence was recorded. Data shown are the mean ± SEM from five independent experiments. ** *p* < 0.01 and *** *p* < 0.001 compared to respective nucleotide alone, analysed using a one-way ANOVA with Bonferroni post hoc test.

**Table 1 ijms-21-08572-t001:** Nucleotide-induced changes in intracellular Ca^2+^ in DH82 cells as measured by half maximal effective concentration.

Nucleotide	Peak Ca^2+^ Response		Net Ca^2+^ Response	
	pEC_50_	Hill Coefficient	pEC_50_	Hill Coefficient
ATP	5.88 ± 0.05 (100%)	0.99	5.92 ± 0.09 (100%)	1.43
UTP	6.16 ± 0.09 (65.9%)	1.02	6.26 ± 0.12 (69.1%)	1.71
ADP ^1^	4.03 ± 0.30 (26.4%) ^2^	1.47	4.07 ± 0.21 (19.9%) ^2^	1.00
BzATP	<4.00 (12.1%) ^3^	2.26	<4.00 (20.5%) ^3^	2.66
UDP	ND (<10%)	-	ND (<10%)	-
2MeSADP	ND (<10%)	-	ND (<10%)	-

Abbreviations: AUC, area under the curve; ND, not determined (pEC_50_ not calculated due to lack of response). Values in parentheses indicate the percent of each maximum agonist response compared to 100 µM ATP.^1^ ADP in the presence of hexokinase to remove contaminating ATP.^2^
*p* < 0.05 compared to the respective pEC_50_ of ATP and UTP (one-way ANOVA).^3^
*p* < 0.01 compared to the respective pEC_50_ of ATP and UTP (one-way ANOVA).

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
