# Peer review of "P2Y2 and P2X4 Receptors Mediate Ca2+ Mobilization in DH82 Canine Macrophage Cells"

_ijms, 2020, doi:10.3390/ijms21228572_

Round 1

Reviewer 1 Report

This MS investigates the ATP/ADP- and UTP/UDP-sensitive receptor-endowment of the DH82 canine macrophage cell-line with the measurement of Ca2+-transients. It is a thorough study performed by the use of adequate techniques. My main criticism relates to the following points: (1) There were no experiments made with native dog macrophages being an important extension of the cell-line investigations; (2) There is also a certain contradiction between the results of semi-quantitative PCR (presence of P2X4-mRNA, but absence of P2Y2-mRNA) and immunohistochemistry (presence of P2Y2-protein). I do not consider it problematic that immunohistochemistry shows major intracellular localization of P2X4, because this fact is well known from the literature, and also mentioned by the authors in the Discussion. (3) The extent of the depression of Ca2+-i by P2X4R antagonists is extremely small. I am missing experiments with the same antagonists on cells pretreated with substances depleting Ca2+-i. In this context the measurement of ATP effects after pre-treatment with e.g. thapsigargin would be quite helpful. Although the authors state that thapsigargin completely abolished the effect of ATP, they show only a combination of EGTA and thapsigargin treatment.

Minor remarks:

  1. There are 7 “mammalian” P2X and 8 “mammalian” P2Y receptors.
  2. It could be mentioned already here that dog P2X4 and P2Y2 have been cloned and the pharmacology of the recombinant receptors have been published.

Reviewer 2 Report

Major comment:

RT-PCR showed high expression of P2X4 receptors and low expression of all P2Y receptors, but immunocytochemistry showed predominant expression of P2Y2 receptors and low levels of  P2X4 receptor. Conclusion (based on calcium measurements) is that  nucleotide-induced responses are due namely to activation of P2Y2 receptors while contribution of P2X4 receptors is relatively small. If it is so, it is not clear why expression of P2Y mRNA is not higher than expression of mRNA for other P2Y receptors or P2X4 receptor. Explanation of this discrepancy is not given and is unconditionally needed.

Minor comments:

Line 32: Why P2Y receptors are written with subscript and P2X not ?

Line 76: Abbreviation ACTB is not explained

Lines 121-122: “...fast decay P2X receptors vs slower sustained decay P2Y receptors…”

Lines 143-145: “Preincubation with 50 µM TNP-ATP partially reduced Ca2+ responses mediated by 10 µM ATP, however, no significant inhibition was observed at 1 µM or 100  µM ATP (Figure 3A-C)”. It means that no concentration dependence was observed? Please explain this unusual observation.

Lines 175-176: ”Preincubation with paroxetine partially reduced Ca2+ responses mediated by ATP concentrations of  10 µM or greater, with no significant inhibitory effect observed at 1 µM or below (Figure 4A-C).” But Figure 4A-C shows that the inhibition of 1 µM  ATP responses is the same as inhibition of 10 µM  or 100 µM  ATP responses.

Line 193: ...movement. To determine…

Lines 224-225: “… there were no significant differences in decay kinetics between cells preincubated in  absence or presence of ivermectin (Figure 6F)”.  It is highly probable that ivermectin-induced prolongation of P2X4 deactivation in DH82 cells could not be observed because deactivation time constant of ATP-induced currents in cells expressing recombinant P2X4 receptors is in  the range of 10-50 s in the presence of ivermectin (Khakh et  al, J Neurosci 1999; Jelinkova et al., BBRC, 2006), while decay time of nucleotide-induced calcium responses in DH82 cells is longer than 50 s even in the absence of ivermectin.

Lines 451-461: Discussion about polymorphisms in canine P2Y2  gene, which primary sequence is not known, is very speculative. Moreover the authors suppose that this polymorphisms  practically does not exist or might be very rare.

Fig. 2: Start and duration of nucleotide application is not shown

Figs. 3, 4 and 5: Panels A-C are not described in text to figure.

Round 2

Reviewer 1 Report

I am not really happy with the authors responses to my queries:

  1. To obtain an animal ethics approval for working with the blood of dogs, should be really simple. I guess you must not write a complete animal protection application but just submit a notification to the relevant authorities. Without experiments with native macrophages the present experiments have very little value.

  1. The authors included an additional Fig. 1 in their response to my question, however, they should include this panel into the relevant Fig. 7.

Round 3

Reviewer 1 Report

In view of the difficulties in obtaining dog blood for macrophage culturing I have no further questions and consider the items raised by me as perfectly clarified.